# OpDetect: A convolutional and recurrent neural network classifier for precise and sensitive operon detection from RNA-seq data

**Rezvan Karaji**[1], **Lourdes Peña-Castillo**[1,2]*

1 Department of Computer Science, Memorial University of Newfoundland, St. John's, Newfoundland and Labrador, Canada, 2 Department of Biology, Memorial University of Newfoundland, St. John's, Newfoundland and Labrador, Canada

* lourdes@mun.ca

**Data availability statement:** Data used are held in a public repository and accession numbers

## Abstract

An operon refers to a group of neighbouring genes belonging to one or more overlapping transcription units that are transcribed in the same direction and have at least one gene in common. Operons are a characteristic of prokaryotic genomes. Identifying which genes belong to the same operon facilitates understanding of gene function and regulation. There are several computational approaches for operon detection; however, many of these computational approaches have been developed for a specific target bacterium or require information only available for a restricted number of bacterial species. Here, we introduce a general method, OpDetect, that directly utilizes RNA-sequencing (RNA-seq) reads as a signal over nucleotide bases in the genome. This representation enabled us to employ a convolutional and recurrent deep neural network architecture which demonstrated superior performance in terms of recall, F1-score and Area under the Receiver-Operating characteristic Curve (AUROC) compared to previous approaches. Additionally, OpDetect showcases species-agnostic capabilities, successfully detecting operons in a wide range of bacterial species and even in *Caenorhabditis elegans*, one of few eukaryotic organisms known to have operons. OpDetect is available at https://github.com/BioinformaticsLabAtMUN/OpDetect.

## Introduction

Bacteria are involved in the survival and functioning of all plants and animals, including humans [1]. Understanding bacterial gene function and regulation is essential to decipher how bacteria interact with other organisms and the environment. Operons, fundamental to bacterial genome organization and gene regulation, play a critical role in bacterial molecular functions. An operon refers to a group of neighbouring genes that are regulated by one or more overlapping transcription units [2]. These overlapping transcription units are transcribed in the same direction and have at least one gene in common. Transcription units are DNA regions that encompass the area from a promoter, where transcription is initiated, to a

are provided within the manuscript. All code written in support of this publication is publicly available at https://github.com/Bioinformatics LabAtMUN/OpDetect and we have archived our code on Zenodo (DOI: 10.5281/zenodo. 15186253).

**Funding:** This research was partially funded by a Natural Sciences and Engineering Research Council of Canada (NSERC, https://www.nserc-crsng.gc.ca/index_eng.asp) Discovery Grant (#2019-05247) to L.P.-C., and a graduate fellowship from Memorial University School of Graduate Studies (www.mun.ca/sgs/) to to R.K. The funders did not play any role in the study design, data collection and analysis, decision to publish or preparation of the manuscript.

**Competing interests:** The authors have declared that no competing interests exist.

terminator, which marks the end of transcription [2]. Thus, genes in a transcription unit are transcribed as a single mRNA. Genes belonging to the same operon are typically, but not necessarily, functionally related or involved in the same metabolic pathways [2–5]. Operon detection is the task of identifying genes belonging to the same operon, which contributes to the mapping of regulatory networks. This will lead to a better understanding of gene functionality and regulation in bacterial genomes [4,6].

Several machine learning-based tools have recently been developed for operon detection in bacterial genomes. These tools utilize various computational approaches. Operon-mapper [7] utilizes computational algorithms grounded in the Prokaryotic Operon Database (ProOpDB) [8,9]. It employs a two-layer neural network that incorporates intergenic distances and gene functional relationship scores from STRING [10]. Its training and evaluation was done in *Escherichia coli* and *Bacillus subtilis*. For RNA-seq data analysis and bacterial operon prediction, Rockhopper [11] adopts a Naïve Bayes model that integrates intergenic distances and gene expression levels from RNA-seq data. The training and evaluation of Rockhopper's model was done on *E. coli*, *B. subtilis*, and *Helicobacter pylori*. Operon Finder [12] is a web service that builds upon Operon Hunter [4]. Operon Hunter introduces a distinct approach by adapting, using transfer learning, ResNet18 [13] on visual comparative genome representations of six species with experimentally confirmed operons in the Operon DataBase (ODB) [14]. Comparative genomic images were obtained via the Compare Region Viewer service from PATRIC [15]. OperonSEQer [16] uses a voting system that combines six machine learning algorithms. Based on RNA-seq reads, OperonSEQer extracts Kruskal-Wallis statistics and p-values to evaluate gene pairs and intergenic regions for differences in read coverage. OperonSEQer is trained on computationally-obtained operon labels from MicrobesOnline [17] for eight different organisms.

While all these computational techniques have significantly advanced operon detection, there are still some opportunities for improvement. First, some of these tools rely on features from external sources, these features are often available only for extensively studied organisms. This affects the applicability of such models to lesser-studied bacterial genomes, which comprise the vast majority of bacteria [18]. Second, some models have been trained on computationally derived labels, potentially leading to mimicry of another algorithm's decisions. Third, some models have been trained and evaluated in the same organisms, which could lead to data leakage. These last two issues might cause that reported performances are overestimated. In this work, we have developed a tool to address these issues. Our approach, OpDetect, relies solely on RNA-seq data, which are available for a broad range of bacteria. It was trained on experimentally verified operon labels, and evaluated on an independent set of organisms not used for training. Additionally, we focus on creating an organism-agnostic operon detection model capable of adapting to different species and effectively identifying operons across various conditions. We comparatively assessed the performance of OpDetect with that of Operon-mapper, OperonFinder, OperonSEQer and Rockhopper. Our results show that OpDetect outperforms these other four approaches in terms of recall, F1-score and AUROC.

## Materials and methods

### Data

This study utilizes genome sequences, RNA-seq data, and operon annotations as primary data sources. Thus, we collected data for organisms that have operon annotations in OperonDB and RNA-seq data publicly available. The genome sequence and annotation files are

obtained from the RefSeq [19] database. RNA-seq data for up to six samples of each organism are sourced from the Sequence Read Archive (SRA) [20] and the European Nucleotide Archive (ENA) [21]. Operon annotations are obtained from OperonDB version 4 (ODB) [14]. This curated database, available at https://operondb.jp/, contains experimentally known operons. Detailed information on organisms and their accession codes for genome sequences and RNA-seq data is available in Table 1. We selected seven bacterial organisms to be used for training purposes and another seven organisms for independent validation (six bacteria and one eukaryote). This selection was based on data availability: Those bacterial organisms with more annotated gene pairs were used for training.

## Data preparation commands

The initial steps in processing the data involve trimming and filtering the raw sequencing data in FASTQ format. This process is performed using Fastp (version 0.23.1) [34]. The trimmed and filtered FastQ files are then aligned to the reference genomes using HISAT2 (version

**Table 1. Data used in this study.**

| Training | | | | | | |
|---|---|---|---|---|---|---|
| **Organism** | | **RNA-seq** | | | | |
| **Name (Phylum)** | **Genome** | **Study** | **Samples** | | | **Ref** |
| *B. subtilis* subsp. subtilis str. 168 (Bacillota) | NC_000964.3 | GSE179533 | SRR15049591 | SRR15049592 | SRR15049593 | [22] |
| | | E-MTAB-10658 | ERR6156944 | ERR6156945 | ERR6156946 | [23] |
| *Corynebacterium glutamicum* ATCC 13032 (Actinomycetota) | NC_006958.1 | GSE120924 | SRR7977557 | SRR7977561 | SRR7977565 | [24] |
| | | E-MTAB-8070 | ERR3380462 | ERR3380465 | ERR3380468 | [25] |
| *E. coli* K-12 substr. MG1655 (Pseudomonadota) | NC_000913.2 | GSE65642 | SRR1787590 | SRR1787592 | SRR1787594 | [26] |
| | | GSE114917 | SRR7217927 | SRR7217928 | SRR7217929 | [27] |
| *Helicobacter pylori* 26695 (Campylobacterota) | GCA_000008525 | GSE94268 | SRR5217496 | | | [28] |
| *Legionella pneumophila* str. Paris (Pseudomonadota) | NC_006368.1 | E-MTAB-4095 | ERR1157043 | ERR1157044 | ERR1157045 | [29] |
| *Listeria monocytogenes* EDG-e (Bacillota) | NC_003210.1 | GSE152295 | SRR11998208 | SRR11998211 | SRR11998214 | [30] |
| | | | SRR11998217 | SRR11998220 | SRR11998223 | |
| *Mycoplasmoides pneumoniae* M129 (Mycoplasmatota) | NC_000912.1 | E-MTAB-8537 | ERR3672190 | ERR3672191 | ERR3672192 | [31] |
| | | | ERR3672193 | | | |
| **Validation** | | | | | | |
| **Organism** | | **RNA-seq** | | | | |
| *Photobacterium profundum* SS9 (Pseudomonadota) | NC_006370.1, NC_006371.1 | GSE38259 | SRR500950 | SRR500951 | | [32] |
| *Agrobacterium fabrum* str. C58 (Pseudomonadota) | NC_003062, NC_003063 | GSE173921 | SRR14432343 | SRR14432344 | SRR14432345 | NA |
| *Borreliella burgdorferi* B31 (Spirochaetota) | NC_001318.1 | GSE152295 | SRR11997800 | SRR11997801 | SRR11997802 | [30] |
| *Bradyrhizobium diazoefficiens* USDA 110 (Pseudomonadota) | NC_004463.1 | GSE163004 | SRR13238987 | SRR13238988 | SRR13238989 | [33] |
| *Pseudomonas aeruginosa* PAO1 (Pseudomonadota) | NC_002516.2 | GSE152295 | SRR11998427 | SRR11998428 | SRR11998429 | [30] |
| *Yersinia pestis* CO92 (Pseudomonadota) | NC_003143.1 | PRJNA384395 | SRR5489122 | SRR5489125 | | NA |
| *C. elegans* (Nematoda) | GCF_000002985.6 | GSE149300 | SRR11605370 | SRR11605378 | SRR11605385 | NA |

2.2.1) [35], and the read coverage for each genome base is extracted using SAMtools (version 1.17) [36] and BEDtools (version 2.30.0) [37]. Complete commands for these steps are provided in OpDetect's GitHub repository.

## Feature representation

The feature representation used in this study draws inspiration from signal processing techniques, particularly the work of [38] in classifying human activities using signals from wearable sensors. However, instead of analyzing signals over time, we focus on the RNA-seq read counts across nucleotide bases in a genome. In our case, the different sensors correspond to multiple RNA-seq samples of the same organism. This perspective allows us to leverage signal processing advancements for our task, operon detection.

By adopting this representation, we aim to maximize the utilization of information from RNA-seq data. Previous approaches such as OperonSEQer often relied on statistical analyses of RNA-seq data, which by summarizing the data with statistics removed potential informative patterns from the input data. Our feature representation enables us not only to use nucleotide-level signals but also to combine them with primary sequence features, such as gene length, gene borders, and intergenic distances. Another advantage of our feature representation is the compatibility of the final dataset's shape with convolutional neural network architectures.

The features in our model are derived from read counts per genome base, paired with operon labels obtained from ODB. The read counts for each gene are grouped based on genome annotations, allowing the extraction of gene-specific vectors. These vectors are then paired for same-strand consecutive genes. Read counts in each pair of genes together with their intergenic region read counts are assembled into a single vector. The process is repeated for up to six samples for each organism. All vectors are resampled to a fixed size of 150 (Fig 1). Following resampling, scaling transforms the read counts to a range of 0 to 1, akin to treating gene pairs as images. Separate channels are constructed for each part of the gene pairs: the first gene, intergenic region (IGR), and the second gene (Fig 1). This approach, reminiscent of RGB channels in images, allows the treatment of each vector as an independent entity within the feature representation. Additionally, this representation naturally encodes the length and borders of each segment (i.e., gene, intergenic region, gene). Visualizations of the features

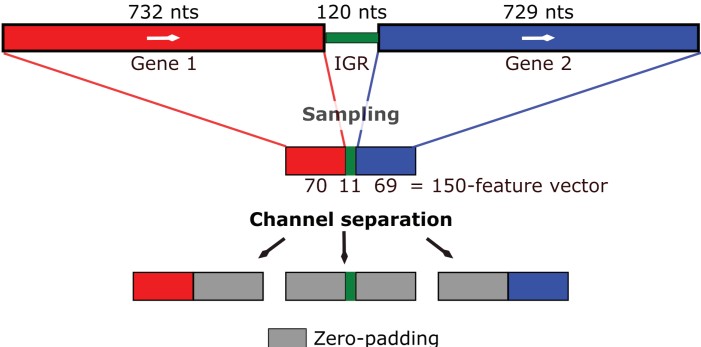

**Fig 1. Feature generation process.** The read counts per nucleotide (nt) for Gene 1, intergenic region (IGR) and Gene 2 are proportionally sampled into a 150-feature vector. This vector is separated into three channels, one for each region. On each separated vector, the features outside the corresponding region are zeroed. If there is not IGR, the corresponding channel will be all zeros.

generated for an operon and a non-operon are provided in S1 File. The final shape of each gene pair's vector is (150, 6, 3), reflecting the fixed size input vector, number of samples, and number of channels. Note that these six vectors contain the only features used to train our model. That is, the genome sequence and annotation are only used to map the reads to the corresponding genome and generate these feature vectors.

Operon labels, assigned based on ODB annotations, distinguish gene pairs within the same operon (label 1) or not (label 0), with label 2 indicating insufficient experimental evidence for operon relationship determination. As ODB contains only experimentally validated operons, we labelled contiguous genes listed in ODB as belonging to different operons as non-operons (label 0). To avoid introducing false negatives in the training data, we introduced label 2 to account for the fact that lack of annotation is not the same as knowing that two genes are not within the same operon. Label 2 is given to pairs of genes absent from the ODB annotations. The reason for this is that if two contiguous genes have been studied and found to be in different operons, then it is more likely that they do not belong to the same operon (these pairs of genes are given label 0) than two contiguous genes for which their operon is unknown (these pairs of genes are given label 2). Gene pairs labeled as 2 are excluded from the training data. The number of instances (gene pairs) for each label in our training data is provided in Table 2.

## Machine learning model

CNN-LSTM is the deep learning architecture we adapted from [38]. It combines Convolutional Neural Networks (CNNs) and Long Short-Term Memory (LSTM) recurrent neural networks [39] to capture spatial and sequential dependencies in the data. The CNN-LSTM architecture that is utilized in this study includes a CNN, a Lambda, an LSTM, an Attention, and a Dense layer, respectively (Table 3). The CNN layer acts as a feature detector, extracting spatial patterns from the input data. A Lambda layer facilitates a smooth transition to the LSTM layer by reshaping the CNN layer output. The LSTM layer captures sequential dependencies, utilizing its memory to predict downstream transcription levels. An Attention layer enhances focus on relevant data regions. To prevent overfitting, dropout regularization is applied. The Dense layer processes information for the final binary output. OpDetect is an ensemble of ten of these CNN-LSTM networks and outputs as the probability of a pair of genes being in the same operon the average probability of this ensemble.

## Software and packages

The feature representation pipeline and machine learning model in this study were developed using Python (version 3.10.13). The list of used packages along with their versions is included in S1 File. OpDetect code is available at https://github.com/BioinformaticsLabAtMUN/OpDetect.

**Table 2. The number of instances (gene pairs) per label class in our training data. Labels 0 and 1 are used to train the model, making the final size of training data 10375 gene pairs.**

| Label | Operon(1) | Non-operon(0) | Unknown(2) |
|---|---|---|---|
| Number of gene pairs | 6345 | 4030 | 9828 |

**Table 3. CNN-LSTM architecture. (None, 150, 6, 3) in the input layer specifies that the CNN model can accept input data with variable batch sizes (None), where each sample has a length of 150 elements (fixed size input vector), a height of 6 rows (number of samples), and a width of 3 columns (number of channels). The Lambda layer is used to adjust the shape of the output from the CNN layer to input for the LSTM layer.**

| Layer | Output Shape | No. parameters |
|---|---|---|
| Input Layer | (None, 150, 6, 3) | 0 |
| Conv2D | (None, 146, 1, 64) | 5,824 |
| Lambda | (None, 146, 64) | 0 |
| LSTM | [(None, 146, 64), (None, 64), (None, 64)] | 33,024 |
| Self Attention | [(None, 1024), (None, 6, 146)] | 2,560 |
| Lambda | (None, 1024) | 0 |
| Dense | (None, 2) | 2,050 |

## Training the model

We trained our model using the data shown in Table 2. To prevent overfitting when optimizing the hyperparameters, 10-fold grid-search cross-validation was employed, averaging results across folds. The hyperparameters listed in Table 4 were selected to achieve a high AUROC. AUROC is a metric used to evaluate the performance of binary classification models. The AUROC can be interpreted as an estimate of the probability that a random positive instance will be predicted to have a higher likelihood to belong to the positive class (operon) than a random negative instance. The specifications for the attention layer are exactly as defined in [38]. We use the default confidence threshold of 0.5 to classify gene pairs as operons. This threshold is applied on the probabilities output by OpDetect. To support the reproducibility of the machine learning method of this study, the machine learning summary table is included in S1 File as per DOME recommendations [40].

## Evaluation metrics

To evaluate our model's effectiveness in operon detection, we consider the F1-score. The F1-score is defined as:

$$\text{F1-score} = \frac{2 \times \text{Precision} \times \text{Recall}}{\text{Precision} + \text{Recall}} \tag{1}$$

where recall is the proportion of positive instances predicted to be positive, and precision is the proportion of predicted positive instances which are actually positive. However, due to limited 0-label instances in five of the seven validation organisms, F1-score calculation can be based on very few (less than 10) negative instances and hence unreliable. To overcome this, we also assessed the performance of our model using recall. In the cases where there were more than 10 negative instances, we also utilized the AUROC, and visualized the ROC curve to observe the trade-off between the True Positive Rate (TPR) or recall, and the False Positive Rate (FPR) at various classification thresholds.

**Table 4. Hyperparameters used in our final CNN-LSTM model.**

| Hyperparameter | Value | Hyperparameter | Value | Hyperparameter | Value |
|---|---|---|---|---|---|
| Batch size | 32 | Epoch | 100 | Dropout rate | 0.3 |
| Kernel size | (5, 6) | CNN filters | 64 | LSTM units | 64 |
| Optimizer | Adam | Learning rate | 0.001 | | |
| Loss function | Categorical cross entropy | Early stopping Metric | AUROC | Early stopping patience | 10 |

## Results and discussion

### Cross-validation results

For the aggregated training data of seven organisms (Table 1, top half), the performance of OpDetect over 10-fold cross-validation is shown in Table 5. Additionally, we evaluated the performance of a CNN architecture with a GlobalMaxPooling2D layer (instead of the LSTM layer) and a CNN-LSTM architecture without an attention layer (results provided in S1 File), these two architectures were outperformed by OpDetect's final architecture. The number of potential alternative network architectures is combinatorial and exploring all of them is outside the scope of this work. Thus, we cannot be certain that OpDetect's final architecture is the optimal one for this task. Further investigation is needed to assert this.

### Comparative assessment

We evaluated the performance of OpDetect on identifying pairs of genes belonging to the same operon on seven organims (Table 1, bottom half) and compared its performance with that of Operon-mapper, OperonFinder, OperonSEQer and Rockhopper. The number of operons and non-operons collected for these seven organisms is provided in Table 6. Collected data are incomplete, as we only used experimentally-confirmed operons in ODB and we implemented a strict approach to label gene pairs as non-operons. In other words, there are actual operons and also non-operons missing from our data. This has implications during the evaluation of the methods; for example, a predicted operon might in fact be an operon (i.e., the prediction is correct) but it will be considered a false positive because that operon has not been experimentally-confirmed. To test the generalizability of OpDetect, we included in our validation data *B. burgdorferi*'s data and *C. elegans*'s data. *B. burgdorferi* belongs to a phylum (Spirochaetota) not present in our training data and *C. elegans* is one of the few eukaryotes known to have operons [41].

During the comparative assessment, we faced some challenges with some of the methods included in the evaluation. These challenges were: i) Due to their size, we were unable to load the genome of *C. elegans* to the Operon-mapper website for prediction. As a result,

**Table 5. OpDetect's 10-fold cross-validation results. The 90% confidence interval suggests that the model's performance metric is 90% probable to be within this range.**

| Performance metric | Mean value | 90% Confidence interval |
|---|---|---|
| Recall | 89.17% | [88.41%, 89.92%] |
| F1-score | 89.71% | [88.98%, 90.43%] |
| Accuracy | 90.37% | [89.70%, 91.04%] |
| AUROC | 0.892 | [0.884, 0.899] |

**Table 6. Number of gene pairs belonging to the same operon (P) and not belonging to the same operon (N) per organism in our validation data.**

| Organism | No. of Operons (P) | No. of Non-operons (N) |
|---|---|---|
| *C. elegans* | 1184 | 56 |
| *P. profundum* | 676 | 87 |
| *P. aeruginosa* | 67 | 3 |
| *B. diazoefficiens* | 15 | 2 |
| *B. burgdorferi* | 20 | 0 |
| *A. fabrum* | 14 | 0 |
| *Y. pestis* | 6 | 0 |

we do not possess predictions of *C. elegans* operons from Operon-mapper. ii) Due to the reliance of Operon Finder on external data sources, the predictions generated by this method are restricted to the organisms that are available within those specific sources. Consequently, we were unable to obtain predictions from Operon Finder for *Y. pestis* CO92, *B. diazoefficiens* USDA 110, and *C. elegans*. iii) OperonSEQer predicted as belonging to the same operon gene pairs from different strands, whereas it is known that genes within the same operon must be located on the same strand [42]. iv) OperonSEQer and Operon Finder did not provide a prediction for each gene pair.

In the comparative assessment when a method failed to make a prediction for a pair of genes, we considered this as predicting that the gene pair was a non-operon and assigned a probability of 0.49 so that the performance metrics were calculated on the same number of gene pairs for all methods. We decided to do this as we argue that failing to make a prediction for a consecutive gene pair on the same strand should be penalized. Assigning a probability of 0.49 indicates that this data point is on the decision boundary of the classifier (i.e., both labels are almost equiprobable) and should have less impact on the calculation of the AUROC than assigning a different probability value. To calculate the AUROC, when possible, we used probabilities. For OperonSEQer, we used as probabilities the average over its six models' predictions. For Operon Finder which predicts a probability per gene instead of per gene pair we used as probability for a gene pair the average of the probabilities of the individual genes. In the case of Rockhopper, we used the predictions (i.e., 0s and 1s) to draw the ROC curves and calculate the AUROCs. The results of our comparative assessment are provided in Table 7. Fig 2 shows the ROC curves for *C. elegans* and *P. profundum*.

These results indicate that OpDetect can reliably identify operons even in an eukaryote (*C. elegans*) while Rockhopper and OperonSEQer, the two other tools which can predict *C. elegans*'s operons, are close to or below the random classifier's performance (Fig 2 left).

**Table 7. F1-scores for the two validation organisms with more than 10 non-operon labels, *C. elegans* and *P. profundum* SS9, and recalls for all seven validation organisms. The highlighted numbers in the tables represent the best performance per organism. The mean recall for Operon-mapper and Operon Finder was calculated excluding the missing values. \*indicates that the target organism is included in the training data of the corresponding method.**

| F1-score | | | | | |
|---|---|---|---|---|---|
| Organism | Operon-mapper | Rockhopper | Operon Finder | OperonSEQer | OpDetect |
| *C. elegans* | NA | 11% | NA | 32% | 79% |
| *P. profundum* SS9 | 58% | 40% | 91%* | 96% | 95% |
| **Recall** | | | | | |
| Organism | Operon-mapper | Rockhopper | Operon Finder | OperonSEQer | OpDetect |
| *C. elegans* | NA | 6% | NA | 19% | 66% |
| *P. profundum* SS9 | 42% | 25% | 89%* | 99% | 96% |
| *P. aeruginosa* PAO1 | 60% | 52% | 94% | 91% | 100% |
| *B. diazoefficiens* USDA 110 | 67% | 20% | NA | 73% | 100% |
| *B. burgdorferi* B31 | 25% | 30% | 85% | 75% | 100% |
| *A. fabrum* str. C58. | 14% | 36% | 93% | 86% | 100% |
| *Y. pestis* CO92 | 100% | 100% | NA | 100% | 100% |
| **Average ± s.d.** | 51.3 ± 31.2 | 38.4 ± 30.6 | 90.2 ± 4.1 | 77.6 ± 27.9 | 94.6 ± 12.7 |

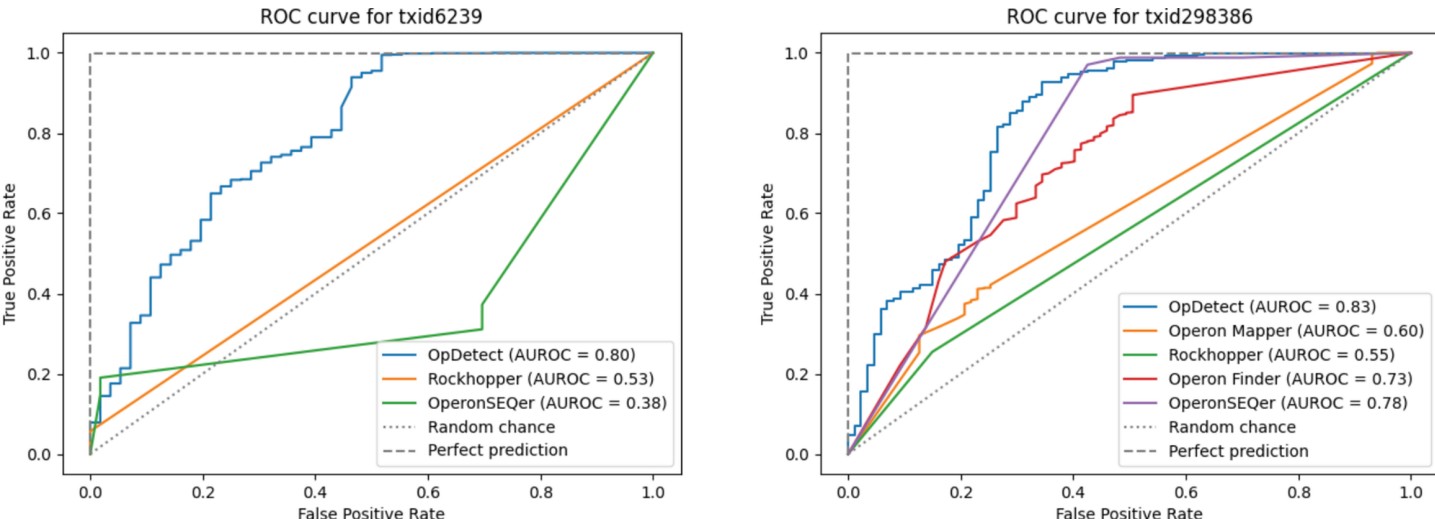

**Fig 2. ROC curve for *C. elegans* (left) and *P. profundum* SS9 (right).** Diagonal dotted line indicates a random classifier and top dashed line indicates a perfect classifier.

For *P. profundum*, the predictive performance of OperonSEQer and OpDetect is comparable, with OperonSEQer achieving the highest F1-score (96% vs 95%, Table 7) and OpDetect obtaining the highest AUROC (0.83 vs 0.78, Fig 2 right). Additionally, OpDetect achieves the highest average recall with the second-smallest standard deviation, suggesting that OpDetect's recall is consistent for a wide range of organisms.

To further explore OpDetect prediction of *C. elegans*'s operons, we generated predictions for all *C. elegans*'s consecutive gene pairs transcribed on the same strand. OpDetect predicts 19.8% of the gene pairs as being part of an operon. This corresponds to 33.8% of *C. elegans* genes predicted as being part of an operon. It has been reported that at least 15 to 17% of *C. elegans* genes are in operons [43,44]. The number of genes predicted by OpDetect as being in an operon is consistent with this lower bound and can be considered an upper bound. This warrants further investigation.

**Assessing the effect of data leakage.** When we performed our comparative assessment, we noticed that, in one case, one of the methods included in the comparative assessment had data from the corresponding validation organism in its training data (Table 7). This is a form of data leakage in the context of claiming that an approach can predict operons in an unseen organism. Data leakage refers to the introduction of information in the training data (about the target function or about the test data) that should not be legitimately available to learn from. Data leakage usually causes an overestimation of a model's predictive performance. Data leakage is a widespread problem in machine learning applications [45], including machine learning-based bioinformatics [46]. Thus, we decided to quantify the effect of having the target organism in the training data. To do this, we evaluated all the methods in the training data and included two versions of OpDetect: i) one where we trained OpDetect and assessed its performance using cross-validation on data that included the target organism (data leakage-affected version). In this version, gene pairs of the target organism were randomly partitioned into training and testing. Thus, individual gene pairs were not in the training and testing data simultaneously; and ii) another version we retrained excluding all gene pairs of the target organism (data leakage-free version). This means that we re-trained OpDetect for each of the training organisms so that we could have a data leakage-free version for

each organism in our training data. Note that excluding the target organism from the training of the other methods was not possible since we used their pre-trained models.

Table 8 presents the F1-scores and AUROCs for each method on the training organisms. "OpDetect NE" refers to OpDetect without excluding the examined organism from the training process (i.e., affected by data leakage). The ROC curves for these organisms can be found in S1 File. For OpDetect, data leakage on average inflates by 2.5% both performance metrics, and this overestimation of predictive performance can be as high as 5%. This highlights the need to ensure a strict separation of training and testing/validation data when assessing the generalizability of machine learning-based models.

OpDetect outperforms all other methods in terms of average F1-score and average AUROC and, considering only data leakage-free methods per organism, OpDetect is the method that achieves the highest F1-score and AUROC in all seven bacteria (Table 8).

**OpDetect achieves stable and accurate operon prediction for a wide range of organisms.** Figs 3 and 4 show the distribution of F1-scores and AUROCs achieved by the five methods in the seven training organisms and the two validation organisms with more than 10 non-operon labels (Tables 1 and 6). For these analyses, we used the OpDetect data leakage-free result (Table 8). OpDetect has less spread in its performance in terms of F1-score and

**Table 8. F1-scores and AUROCs achieved on the training organisms.** * indicates that the target organism is included in the training data of the corresponding method (i.e., data leakage has occurred)."OpDetect NE" refers to OpDetect without excluding the examined organism from the training process (i.e., affected by data leakage). The highlighted numbers in the tables represent the best performance per organism achieved without data leakage.

**F1-scores**

| Organism | Operon-mapper | Rockhopper | Operon Finder | OperonSEQer | OpDetect | OpDetect NE |
|---|---|---|---|---|---|---|
| E. coli | 49%* | 73%* | 86%* | 88%* | 85% | 90%* |
| C. glutamicum | 50% | 12% | 79%* | 81% | 86% | 89%* |
| L. monocytogenes | 57% | 20% | 92%* | 65% | 97% | 98%* |
| L. pneumophila | 56% | 28% | 86%* | 89% | 93% | 96%* |
| H. pylori | 59% | 14%* | 90% | 93% | 97% | 97%* |
| B. subtilis | 57%* | 74%* | 90%* | 87%* | 93% | 95%* |
| M. pneumoniae | 63% | 42% | 48% | 87% | 89% | 90%* |
| **Average ± s.d.** | 56 ± 5% | 38 ± 26% | 82 ± 15% | 84 ± 9% | 91 ± 5% | 94 ± 4% |

**AUROC**

| Organism | Operon-mapper | Rockhopper | Operon Finder | OperonSEQer | OpDetect | OpDetect NE |
|---|---|---|---|---|---|---|
| E. coli | 0.68* | 0.78* | 0.91* | 0.88* | 0.95 | 0.97* |
| C. glutamicum | 0.65 | 0.53 | 0.82* | 0.77 | 0.92 | 0.95* |
| L. monocytogenes | 0.74 | 0.56 | 0.90* | 0.69 | 0.97 | 0.98* |
| L. pneumophila | 0.66 | 0.57 | 0.79* | 0.78 | 0.86 | 0.87* |
| H. pylori | 0.74 | 0.54* | 0.89 | 0.86 | 0.97 | 0.98* |
| B. subtilis | 0.73* | 0.79* | 0.89* | 0.86* | 0.94 | 0.97* |
| M. pneumoniae | 0.72 | 0.62 | 0.64 | 0.80 | 0.90 | 0.92* |
| **Average ± s.d.** | 0.70 ± 0.04 | 0.63 ± 0.11 | 0.83 ± 0.1 | 0.81 ± 0.07 | 0.93 ± 0.04 | 0.95 ± 0.04 |

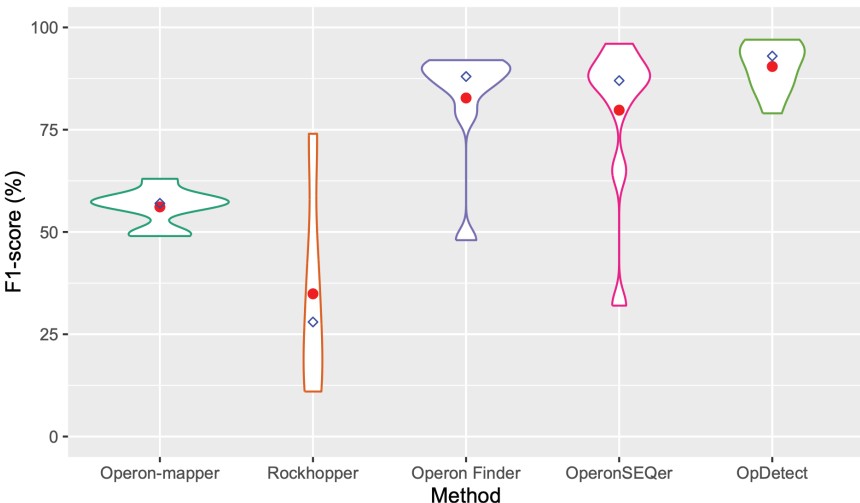

**Fig 3. Violin plots of the F1-scores achieved by each method in the seven training organisms and the two validation organisms with more than 10 non-operon labels.** We used OpDetect data leakage-free result (Table 8). For each method, calculations excluded their missing results. That is, for Operon-mapper and Operon Finder the distribution is on eight organisms instead of nine, as these two methods could not predict operons in *C. elegans*. Red circles indicate the average F1-score and blue diamonds the median F1-score per method.

AUROC than the other methods (Figs 3 and 4) while achieving higher F1-score and AUROC. This indicates that OpDetect is able to balance precision and recall. Although Operon Finder showed the second-best performance in terms of AUROC and F1-score overall (Figs 3 and 4), it is limited to making predictions for specific species due to its reliance on features obtained from external data sources. For example, Operon Finder could predict only four out of seven species in our validation data (Table 7).

We employed the Friedman non-parametric statistical test that is suitable for comparing multiple classifiers over multiple datasets to assess the statistical significance of AUROC differences between the methods [47]. To do this, we ranked each model based on the AUROC obtained on all seven training organisms and two validation organisms, *C. elegans* and *P. profundum* SS9. The model with the highest AUROC was assigned rank one, while ties received the same rank. Missing values were ranked last. The Friedman test yielded a p-value of $1.89 \times 10^{-5}$, indicating that the mean AUROC rank obtained by certain classifiers significantly deviates from the others. To find out which models differ in terms of AUROC, we used several pairwise post hoc tests as recommended in [48]; namely, Quade, Miller, Nemenyi, and Siegel post hoc tests. All statistical tests were carried out in R (version 4.4.2) using the packages PMCMRplus (version 1.9.12) [49] and scmamp (version 0.3.2) [50]. We observed that two groups of classifiers with similar ranks emerged, as illustrated in Fig 5. OpDetect's ranks are comparable to those of Operon Finder and OperonSEQer. All pairwise post hoc tests agreed that OpDetect's AUROC ranks are statistically better (adjusted p-values <0.002) than those of Operon-mapper and Rockhopper. There were no statistically differences consistently found among the AUROC ranks of the other four methods.

## Case study: Detecting noncontiguous operons

Noncontiguos operons (NcOs) [51] are a special case of operons where genes belonging to the same transcription unit (i.e., co-transcribed) are separated by an antisense gene. Iturbe et al.

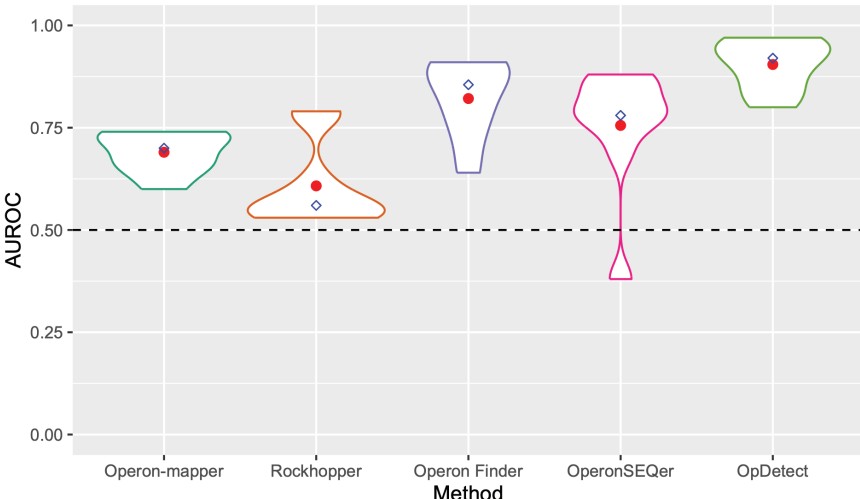

**Fig 4. Violin plots of the AUROC achieved by each method in the seven training organisms and the two validation organisms with more than 10 non-operon labels.** We used OpDetect data leakage-free result (Table 8). For each method, calculations excluded their missing results. That is, for Operon-mapper and Operon Finder the distribution is on eight organisms instead of nine, as these two methods could not predict operons in *C. elegans*. Red circles indicate the average AUROC and blue diamonds the median AUROC per method. Horizontal dashed line indicates the AUROC of a random classifier.

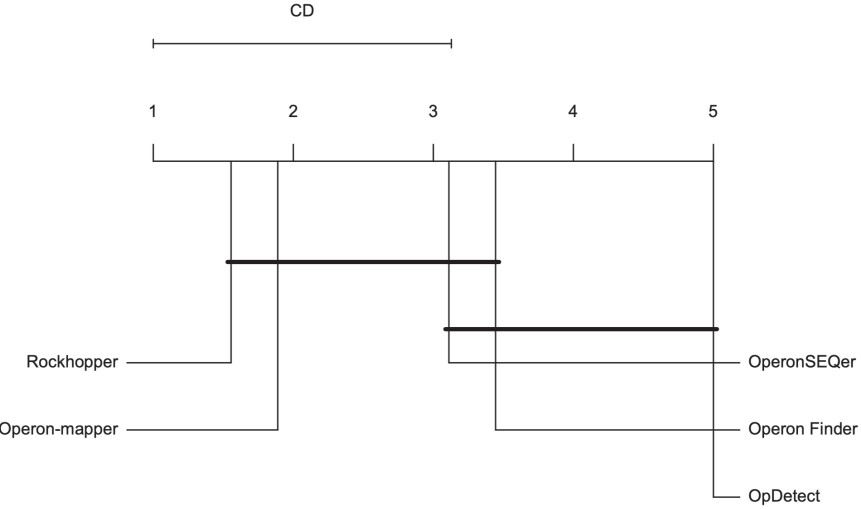

**Fig 5. Critical Difference plot for AUROC, over seven training organisms and two validation organisms.** Classifiers that do not show significant differences according to the Nemenyi test at a significance level of 0.05 are connected with a horizontal line. The methods with the best performance are to the right.

(2024) [52] stated that NcOs were not detected by then-current operon identification tools, and showed how 18 NcOs in *Staphylococcus aureus* could be detected by Nanopore direct RNA sequencing technology. We confirmed that OperonSEQer and Operon-mapper did not predict any of these NcOs. Thus, we decided to further test OpDetect on this challenge. To do this, we used the chromosome sequence and genome annotation of *Staphylococcus aureus* subsp. aureus NCTC 8325 (NC_007795.1), and Illumina NovaSeq 6000 RNA-seq data for this

bacterium (GEO accession ID GSE265954, "WT No calprotectin" samples). In these 18 NcOs, there were 49 pairs of genes and 18 of these 49 gene pairs jumped over an antisense gene (see Figure 1B of [52]). Out of the 31 pairs of genes separated by an intergenic region (instead of an antisense gene), OpDetect identified 28 (or 90%) as belonging to the same operon. The three gene pairs not identified have a predicted probability of being an operon of 0.2, 0.41 and 0.46. Out of the 18 gene pairs separated by an antisense gene (Table 9), OpDetect identified five (or 28%). Two more of these gene pairs have a probability of being an operon greater than 0.2. As future work, it might be worth optimizing the confidence threshold instead of using the default of 0.5.

Additionally, we looked at whether there were other gene pairs (not detected in [52]) predicted by OpDetect as belonging to the same operon that are separated by an antisense gene. There were eleven such gene pairs (Table 10). Further research is needed to confirm these gene pairs as being in a NcOs. The results of this case study indicate that OpDetect can identify NcOs, opening the door to the computational identification of this special type of operons.

## Conclusion

In this work, we introduce OpDetect, a species-agnostic method for operon identification from RNA-seq data. OpDetect is, in fact, the only method we evaluated capable of precise

**Table 9. Eighteen *S. aureus* gene pairs identified in [52] as belonging to the same operon while being separated by an antisense gene. As criteria to deem these gene pairs NcOs, Iturbide et al. have the restriction that the average read depth coverage at three consecutive nucleotides in the intergenic region (i.e., the intervening antisense gene) does not fall below 0.1 reads per nucleotide. The SAOUHSC_ preceding the NCBI gene symbols is omitted in the table. Highlighted are those gene pairs predicted by OpDetect as belonging to the same operon with a probability of at least 0.2.**

| Gene pair | Strand | OpDetect's probability | Max. probability in ensemble |
|---|---|---|---|
| 00208 - 00211 | → | 0.01 | 0.02 |
| 00472 - 00474 | → | 0.87 | 0.94 |
| 00693 - 00695 | → | 0.01 | 0.02 |
| 00825 - 00827 | → | 0.03 | 0.10 |
| 00979 - 00981 | → | 0.00 | 0.01 |
| 01073 - 01071 | ← | 0.23 | 0.51 |
| 01072 - 01074 | → | 0.42 | 0.66 |
| 01199 (fabB) - acpP | → | 0.19 | 0.32 |
| 01330 - 01332 | → | 0.92 | 0.96 |
| 01352 - 01354 | → | 0.59 | 0.75 |
| 01915 (menC) - 01913 (ytkD) | ← | 0.13 | 0.39 |
| 02109 - 02107 | ← | 0.00 | 0.01 |
| 02379 - 02377 | ← | 0.71 | 0.92 |
| 02529 - 02527 (fmhB) | ← | 0.18 | 0.37 |
| 02534 - 02532 | ← | 0.05 | 0.12 |
| 02544 (moaB) - 02542 (moeA) | ← | 0.77 | 0.93 |
| 02833 (strA) - 02836 | ← | 0.02 | 0.04 |
| 02995 - 02991 | ← | 0.00 | 0.00 |

**Table 10. Eleven *S. aureus* gene pairs predicted by OpDetect as belonging to the same operon while being separated by an antisense gene. The SAOUHSC_ preceding the NCBI gene symbols is omitted in the table. These genes were not identified in [52].**

| Gene pair | Strand | OpDetect's probability | Max. probability in ensemble |
|---|---|---|---|
| 00025 - 00027 | → | 0.53 | 0.78 |
| 00287 - 00290 | → | 0.56 | 0.82 |
| 00912 - 00914 | → | 0.51 | 0.75 |
| 02232 - 02235 | → | 0.50 | 0.75 |
| 02658 - 02656 | ← | 0.67 | 0.84 |
| 01938 - 01936 | ← | 0.85 | 0.96 |
| 01583 - 01580 | ← | 0.50 | 0.67 |
| 01490 - 01488 | ← | 0.54 | 0.75 |
| 01447 - 01441 | ← | 0.69 | 0.90 |
| 00155 - 00153 | ← | 0.86 | 0.99 |
| 00028 - 00026 | ← | 0.61 | 0.90 |

and sensitive identification of operons in an eukaryote, *C. elegans*. OpDetect has the following advantages: i) By exclusively relying on genome annotations and RNA-seq data, OpDetect makes operon detection accessible for more species. ii) By using a new feature representation for RNA-seq data, OpDetect can use a CNN-LSTM architecture. This architecture captures both spatial and sequential patterns in the RNA-seq data. iii) By using experimentally evaluated operons and conservatively labelling non-operons, we reduced the number of potentially mislabeled gene pairs in OpDetect training data. It is well-known in machine learning that the cleaner the training data the better the generated model. iv) OpDetect outperforms four state-of-the-art operon prediction methods in terms of F1-score and AUROC. For future work, we propose exploring the inclusion of promoter and terminator data, as they have been reported to have a significant influence on operon detection [5]. Finally, based on our results on assessing the effect of data leakage, we recommend having a strict partition of organisms for training and validation of machine learning-based methods to avoid overestimating their generalizability.

## Supporting information

**S1 File. Supporting information**.
(PDF)

## Acknowledgments

We thank Gavin Hull for his assistance in optimizing OpDetect's hyperparameters. This research was partially enabled by the computing infrastructure provided by Acenet (acenet.ca) and the Digital Research Alliance of Canada (alliancecan.ca).

## Author contributions

**Conceptualization:** Lourdes Peña-Castillo.

**Data curation:** Rezvan Karaji.

**Formal analysis:** Rezvan Karaji.

**Funding acquisition:** Lourdes Peña-Castillo.

**Methodology:** Lourdes Peña-Castillo.

**Software:** Rezvan Karaji.

**Supervision:** Lourdes Peña-Castillo.

**Validation:** Rezvan Karaji.

**Writing – original draft:** Rezvan Karaji, Lourdes Peña-Castillo.

**Writing – review & editing:** Lourdes Peña-Castillo.

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
