## [Decision Letter · Decision Letter 0]

6 Jun 2025

PONE-D-25-24402OpDetect: A convolutional and recurrent neural network classifier for precise and sensitive operon detection from RNA-seq dataPLOS ONE

Dear Dr. Peña-Castillo,

Thank you for submitting your manuscript to PLOS ONE. After careful consideration, we feel that it has merit but does not fully meet PLOS ONE’s publication criteria as it currently stands. Therefore, we invite you to submit a revised version of the manuscript that addresses the points raised during the review process.

We'are sorry for the delay with the review. 

We look forward to receiving your revised manuscript.

Kind regards,

Ivan S Petrushin, Ph.D

Academic Editor

PLOS ONE

Reviewers' comments:

Reviewer's Responses to Questions

**Comments to the Author**

1. Is the manuscript technically sound, and do the data support the conclusions?

Reviewer #1: Partly

Reviewer #2: Yes

2. Has the statistical analysis been performed appropriately and rigorously? 

Reviewer #1: No

Reviewer #2: Yes

3. Have the authors made all data underlying the findings in their manuscript fully available?

Reviewer #1: Yes

Reviewer #2: Yes

4. Is the manuscript presented in an intelligible fashion and written in standard English?

Reviewer #1: Yes

Reviewer #2: Yes

5. Review Comments to the Author

Reviewer #1: Review of OpDetect: A convolutional and recurrent neural network classifier for precise and sensitive operon detection from RNA-seq data

Summary

The authors develop OpDetect, a deep learning approach for detecting operons in a speices-agnostic manner using RNA-sequencing (RNA-seq data). They use only genome sequences, RNA-seq data and existing operon annotation, and no additional information. They treat RNA-seq as you would signal data, and process it as such to maximize information utilization. They demonstrate that their approach outperforms state-of-the-art algorithms on recall, F1 and AUROC.

Overall, there is always the need to continually improve predictive biology programs as new data and cutting-edge algorithms come into play, and operon identification, particularly in understudied species, still requires significant research, so this work is relevant to microbiologists and computational biologists, and addresses an area of need. While the approach is distinct from anything that has been attempted and produces high metrics, a few fundamental concerns need to be addressed before publication can be recommended. Additional details are provided below, but in summary, first the authors need to justify their use of their methodology more thoroughly. Second, the metrics used are skewed by their very low numbers of ‘non-operon’ genes, and evaluation on such a unbalanced dataset can affect these metrics significantly. Third, the use of C. elegans data has to be justified with the biology of C. elegans operons in mind, which could change the interpretation of the data significantly.

Major points

1) The combination of two architectures to properly capture the complexity of RNA-seq data is appreciated. However, the justification is not clear. For example, did the authors try a CNN or LSTM alone and find reduced performance? The inclusion of an attention layer is interesting, but it seems like there is a missed opportunity for cross-attention or dual-attention here when asking about the relationship of RNA-seq data between two genes and an intergenic region – why the choice of self attention? Also, is the LSTM required or would more attention layers perform the same function?

2) The small set of non-operons is a potential issue, both with training and testing. I understand the motivation behind only wanting to use experimentally validated non-operons (I assume this is the case? The authors mention the reverse case of a predicted operon being missed, but I assume the reason there are so few non-operons is because most of them are not experimentally validated to the satisfaction of the authors?). The validity of the F1 score will indeed be impacted by this as mentioned, and while using AUROC and recall are good options, they will not tell you the extent to which non-operon calls are valid with this algorithm. It would be worth considering testing performance on putative non-operons, just to demonstrate the broader applicability of the algorithm outside of confirmed non-operons.

3) The observation with C. elegans operon identification is very interesting but raises a number of concerns that the authors need to address. C. elegans operons are fundamentally biologically different from prokaryotic ones. While prokaryotic polycistronic mRNAs are not processed, but rather the genes on the mRNA are individually translated, the operon mRNAs from C. elegans are processed co-transcriptionally before translation. The other methods evaluated in this study are focused on prokaryotic operons, not eukaryotic ones, and are tailored towards detecting those. This then leads to the question of how stably detectable C. elegans operon RNAs are, and whether the RNA preparation involved any selection (eg. polyA, which would likely preclude these intermediate RNAs, or ribo-depletion, which might preserve them though they would remain transient, or neither, which would call into question the quality of the dataset altogether). The C. elegans dataset referenced does not have this level of detail, and does not have an associated study, so clarification is needed here. As such, I wonder if another non-C.elegans eurkaryotic dataset would predict operons? And given the scarcity of non-operons and the methods used to evaluate, it’s very difficult to tell whether these predictions are valid. So before claiming that this method can find the C. elegans operons, I would do a lot of due diligence with respect to:

• How the RNA from C.elegans was collected and processed, and as such are you likely to find these polycistronic mRNAs in the sample?

• Does the algorithm predict operons in higher organisms known not to have operons?

4) The authors do not differentiate between data leakage and using the same organism for training and testing. While there is validity in training on a set of organisms and testing on a different, unseen organism, using the same organism but different datasets for training and testing is not data leakage or cross-validation. The authors should address this discrepancy and re-evaluate their assessments in the data leakage section.

Minor points

1) All vectors are re-sampled to a fixed size of 150. While this is a technique used with image data, the reconstruction fidelity in RNA-seq data is not clear. Visualization of this resampled data, or some other quantitative metric to assess this would be helpful.

2) The inclusion of a label to account for lack of annotation is interesting, but redundant (i.e. how is this different from just removing unannotated data?).

3) Line 141 – I do not understand the meaning of this threshold (and it is not immediately obvious from skimming the referenced paper).

4) Table 3 – why use lambda layers instead of another convolution (first lambda) or fully connected linear layer (second lambda)? Or is that lambda layer a wrapper for just that? A justification would be helpful.

5) Table 7 – recall is an incomplete metric. I would include precision, specificity, F1 (or a subset at least) so the reader can get a better picture of overall performance.

6) All figures were very low resolution and difficult to read.

7) The observation about non-contiguous genes within an operon is interesting, but not enough discussion is there on why existing programs do not identify these and OpDetect does. It is likely due to the requirement for two genes to be adjacent, not a performance issue of existing programs. If this is the case, it should be clarified. Also, with the gene pairs in Table 9, there should be a column stating whether the intervening antisense gene is expressed or not (i.e. is there detectable signal in the intergenic region?).

Reviewer #2: The paper titled "OpDetect: A Convolutional and Recurrent Neural Network Classifier for Precise and Sensitive Operon Detection from RNA-seq Data" presents a novel, species-agnostic deep learning approach for operon detection using RNA-seq data and sequence information.

However, a number of questions arise:

1)Could the authors clarify how they construct the gene pair vector in cases where the intergenic region is smaller than 150 base pairs?

2)While the architecture proposed by the authors is highly parameter-efficient and its hyperparameters are well justified, it remains an open question whether this architecture is truly optimal. Could the authors conduct an ablation study to show that modifying the number of CNN layers or removing certain components of the neural network leads to worse performance?

3)I suggest that the authors report additional metrics on the set of operons that all methods were able to predict. The procedure described by the authors—where “if a method failed to make a prediction for a gene pair, we considered this as predicting that the gene pair was a non-operon and assigned a probability of 0.49”—may influence the comparison results. Clarification on how this assumption affects the evaluation would be valuable.

4)The reformulation of the operon detection problem as a binary classification task is somewhat limiting, as it hinders the ability to capture more complex scenarios—such as when a model fails to predict some intermediate genes within a multi-gene operon, thereby compromising the full operon prediction. Could the authors comment on their choice of metric and perhaps provide counterarguments to this concern?

5)It would also be interesting to conduct an ablation study on the features provided to the network. Is RNA-seq data alone sufficient for accurate operon prediction, or does the nucleotide sequence contribute significantly to model performance?

6)If the nucleotide sequence plays an important role in prediction, could the authors investigate which parts of the sequence the model considers most informative? Are there specific motifs that the model relies on for prediction? This would be especially intriguing given the model’s ability to generalize to previously unseen eukaryotic species.

6. PLOS authors have the option to publish the peer review history of their article (what does this mean?). If published, this will include your full peer review and any attached files.

Reviewer #1: No

Reviewer #2: **Yes: **Dr. Penzar Dmitry

---

## [Author Response · Author response to Decision Letter 1]

9 Jul 2025

Our response to reviewers has been uploaded as a separate file.

---

## [Editor Report · Decision Letter 1]

16 Jul 2025

OpDetect: A convolutional and recurrent neural network classifier for precise and sensitive operon detection from RNA-seq data

PONE-D-25-24402R1

Dear Dr. Peña-Castillo,

We’re pleased to inform you that your manuscript has been judged scientifically suitable for publication and will be formally accepted for publication once it meets all outstanding technical requirements.

Sincerely,

Ivan S Petrushin, Ph.D

Academic Editor

PLOS ONE

Additional Editor Comments:

Please check, that all references contain the DOI or URL if available.

---

## [Editor Report · Acceptance letter]

PONE-D-25-24402R1

PLOS ONE

Dear Dr. Peña-Castillo,

I'm pleased to inform you that your manuscript has been deemed suitable for publication in PLOS ONE. Congratulations! Your manuscript is now being handed over to our production team.

Kind regards,

on behalf of

Dr. Ivan S Petrushin

Academic Editor

PLOS ONE